# Taiwanese Vegetarians Are Associated with Lower Dementia Risk: A Prospective Cohort Study

**DOI:** 10.3390/nu14030588

**Published:** 2022-01-28

**Authors:** Jui-Hsiu Tsai, Ching-Feng Huang, Ming-Nan Lin, Chiao-Erh Chang, Chia-Chen Chang, Chin-Lon Lin

**Affiliations:** 1Department of Psychiatry, Dalin Tzu Chi Hospital, Buddhist Tzu Chi Medical Foundation, Chiayi 622, Taiwan; faanvangogh@gmail.com; 2College of Medicine, Tzu Chi University, Hualien 970, Taiwan; luccfh@gmail.com (C.-F.H.); mingnan.lin@gmail.com (M.-N.L.); justenjoy555@gmail.com (C.-C.C.); 3Program in Environmental and Occupation Medicine, (Taiwan) National Health Research Institutes and Kaohsiung Medical University, Kaohsiung 807, Taiwan; 4Department of Neurology/Neuromedical Scientific Center, Taichung Tzu Chi Hospital, Buddhist Tzu Chi Medical Foundation, Taichung 427, Taiwan; 5Department of Family Medicine, Dalin Tzu Chi Hospital, Buddhist Tzu Chi Medical Foundation, Chiayi 622, Taiwan; 6Institute of Epidemiology and Preventive Medication, College of Public Health, National Taiwan University, Taipei 106, Taiwan; d03849010@ntu.edu.tw; 7Department of Medical Research, Buddhist Tzu Chi Medical Foundation, Hualien 970, Taiwan; 8Department of Cardiology, Dalin Tzu Chi Hospital, Buddhist Tzu Chi Medical Foundation, Chiayi 622, Taiwan

**Keywords:** vegetarian diet, dementia incidence, Alzheimer’s disease, cardiometabolic risk factors

## Abstract

The number of people living with dementia globally is increasing rapidly, and there is no effective therapy. Dietary pattern is one important risk factor for the development and progression of dementia. We undertake this study to determine whether Taiwanese vegetarian diet in midlife affects dementia incidence in later years in a prospective cohort. We followed 5710 participants (average age less than 60) in the Tzu Chi Vegetarian Study (TCVS). We started recruiting in 2005 and followed until the end of 2014 when the database changed from ICD-9-CM to ICD-10-CM codes. The incidence of dementia was obtained through linkage to the National Health Insurance Research Database. We used Cox proportional hazards regression to estimate the hazard ratio of dementia between vegetarians and nonvegetarians. There were 121 cases of dementia (37 vegetarians and 84 nonvegetarians) diagnosed. Vegetarians were associated with reduced risk of clinically overt dementia compared with nonvegetarians (hazard ratio = 0.671, confidence interval: 0.452–0.996, *p* < 0.05) after adjusting for gender, age, smoking, drinking, education level, marriage, regular exercise, and comorbidities with stepwise regression.

## 1. Introduction

According to Alzheimer’s Disease International (ADI), the number of people living with dementia is increasing rapidly globally and is likely to reach 152 million by 2050 [1]. There are many subtypes of dementia, such as vascular dementia, Alzheimer’s disease (AD), mixed dementia with features of more than one cause, and dementia with Lewy bodies. Less common causes include dementias associated with alcohol abuse, brain injury, and infections, and frontotemporal degeneration [2]. There are twelve risk factors listed for dementia, and these risk factors together account for a sizable portion (about 40%) of dementias worldwide, which theoretically could be delayed or prevented [3]. Because effective treatment for dementia is not yet available, the identification and modification of risk factors is a very important strategy in reducing the incidence of dementia [1]. Diabetes, neuropsychiatric symptoms, and diet have been listed as the three most important modifiable risk factors for dementia [3]. The Rotterdam Study attributed higher risk of dementia to increased intake of cholesterol, saturated fat, total fat, and cholesterol in diets and negatively correlated with fish consumption [4]. Mediterranean diets (high proportion of legumes, fruit, vegetables, olive oil; moderate amount of fish; low meat and dairy products intake) had been demonstrated to be protective against dementia in Western countries [5]. In Asia, the Hisayama Study showed that diets with high amount of soybean products, dairy products, and vegetables and low amount of rice appeared to lower dementia risk in elderly Japanese [6]. Similar to the above findings, a cross-sectional survey of dementia in Taiwan demonstrated that the consumption of fish, vegetables, tea, and coffee has potential benefits against dementia [7]. 

Vegetarian diets have been demonstrated to have significant cardiovascular and metabolic benefits, improving weight and glycemic control, decreasing blood pressure and blood lipids, and reversing atherosclerosis [8]. Our previous study also demonstrated negative correlation between a Taiwanese vegetarian diet and cardiometabolic risk factors such as atherosclerosis, diabetes, strokes, and depression [9,10,11], and all these cardiometabolic risk factors predispose to dementia,

With this prospective cohort study, we aim to study the impact of a Taiwanese vegetarian diet in midlife (mostly ovo-lactovegetarian, meatless) on the subsequent development of dementia, and to explore the association with other risk factors.

## 2. Materials and Methods

### 2.1. Research Design

Starting in 2005, we recruited Buddhist Tzu Chi Foundation volunteers (The Tzu Chi Vegetarian Study, TCVS). There were a total of 12,026 participants in the study. These participants were Buddhists and had trained for at least two years with recommendations of alcohol abstinence and smoking cessation before becoming recognized as Tzu Chi volunteers. Because Buddhism advocates compassion toward animals and conserving the environment, volunteers were recommended to become vegetarians as much as possible. 

All participants of TCVS filled out questionnaires for collecting basic social demographics, medical history, and lifestyle factors (alcohol drinking, cigarette smoking, physical activities, etc.), and dietary pattern. The participants who had eaten no meat, fish, or poultry for at least a year were classified as vegetarian. The rest of participants were classified as nonvegetarian. Our study protocol complied with existing regulations, and the Institutional Review Board (IRB) of Dalin Tzuchi Hospital approved the protocol (1 October 2016. # B10503019). A written informed consent was obtained from each participant.

Comorbidities such as hypertension, hyperlipidemia, diabetes, chronic obstructive pulmonary disease, hyperthyroidism, hypothyroidism, cerebrovascular disease, ischemic heart disease, renal disease, and anxiety were ascertained using the International Classification of Diseases, 9th Revision (ICD-9) codes in the National Health Insurance (NHI) database of Taiwan. We followed participants until 31 December 2014 when NHI changed its coding system. The average follow-up duration was 9.2 years. The follow-up is complete since the NHI covered near 100% of Taiwan’s population as of the end of 2014 [12].

### 2.2. Criteria for Exclusion

The number of participants excluded at each step of exclusion in the current study is shown in Figure 1. Individuals not insured with the National Health Insurance (NHI) or with inaccurate ID were excluded. We also excluded participants aged under 50. Moreover, we excluded all participants with dementia or mild cognitive impairment before enrollment and within two years after participating in this study. Participants with missing data in the study questionnaire were also excluded. 

### 2.3. Dementia Ascertainment

Under official legal permission, using the unique identification number, baseline demographics were connected to the National Death Registry and the Research Database of NHI at the Health and Welfare Data Science Center (HWDSC) at the Ministry of Health of Taiwan. To guard the confidentiality of participants, we performed analyses only in the HWDSC, retrieved only summarized data sets. We follow participants until the date of diagnosing dementia or mild cognitive impairments, death and termination of enrollment in the NHI program.

Incident cases of dementia or mild cognitive impairment were defined as those who had diagnosis code of 290.0 (dementia), 290.1 (presenile dementia), 290.4 (vascular dementia), 331.0 (Alzheimer’s disease), 331.1 (frontotemporal dementia), 331.82 (dementia with Lewy bodies), or 331.83 (mild cognitive impairment) in the Clinical Modification, International Classification of Diseases, 9th Revision (ICD-9-CM). The criterion of incidence cases was identified as participants who had outpatient visits twice or at least one inpatient admission to the Neurology or Psychiatry Department with above diagnosis.

### 2.4. Statistical Analysis

We compared demographic characteristics, lifestyle (cigarette smoking, alcohol drinking, and physical activities), and medical comorbidities (hypertension, cerebrovascular disease, diabetes mellitus, and hyperlipidemia) between vegetarians and nonvegetarians utilizing chi-square test and Student’s *t*-test for continuous or categorical variables, respectively. 

We calculated the incidence rate (IR) by dividing numbers of participant with dementia by the person-years of two groups. Follow-up periods were computed from study entry day until the date of diagnosing dementia or mild cognitive impairments, termination of enrollment in the NHI program, or death. A multivariate Cox proportional hazard model was used to calculate hazard ratios (HR) between vegetarian and nonvegetarian groups, with adjusting for age, gender, educational levels, marital status, lifestyle, and vascular comorbidities. Before applying the proportional hazard model, we confirmed compliance with the assumption through test of interaction between each variable. 

We further used demographic characteristics when conducting subgroup analyses. We used a two-sided *p*-value of <0.05 as significant statistically, and SAS v9.4 (SAS Institute, Cary, NC, USA) was used for all our statistical analyses.

## 3. Results

The sociodemographic characteristics of nonvegetarian and vegetarians at baseline are shown in Table 1. The average age of participants at the time of enrollment in both groups was under 60. Vegetarians were less educated, more likely to be female, single/divorced/widowed, and drank alcohol or smoked less habitually. Vegetarians had significantly less diabetes, cerebrovascular disease, and substance use disorder, as shown in Table 2. Vegetarians also had lower prevalence of hypertension, hyperlipidemia, ischemic heart diseases, and major depressive disorders, but this was not statistically significant. 

There were 121 cases of dementia (37 vegetarians and 84 nonvegetarians) identified in the study period, and vegetarians were associated with a significantly reduced risk of dementia. Stepwise hazard ratio estimation of dementia risk in vegetarian compared with nonvegetarian was 0.671 (confidence interval: 0.452–0.996, *p* < 0.05) after adjusting for gender, age, smoking, drinking, education level, marriage, regular exercise, and comorbidities with stepwise regression as shown in Table 3. Figure 2 shows cumulative incidence of dementia for vegetarians and nonvegetarians as Kaplan–Meier survival plot (log-rank *p* = 0.245). 

Stratification by age groups is shown in Table 4. The adjusted hazard ratio for dementia was 0.6 for those less than 75 years of age, but this was not statistically significant, due to small case numbers.

## 4. Discussion

The main finding of this prospective study is that Taiwanese vegetarians in midlife were associated with a significantly lower risk of developing subsequent dementia in later years compared to nonvegetarians (hazard ratio of 0.671, confidence interval: 0.452–0.996, *p* < 0.05) after adjustment (Table 3), but further stratification analysis by age subgroups revealed no statistically significant association because of small case numbers (Table 4). The incidence rate of dementia in our study (nonvegetarians: 2.91 and vegetarians: 2.32 per 1000 person-years) is lower than that in other studies because only those with clinically overt dementia that required medical attention were included in our study. Dementia generally develops in older age, rapidly increasing in incidence after age 65 years, and the majority (about 80%) occurs in people above 75 years of age [2,13,14,15]. The mean age of our cohort at the time of enrollment was less than 60 years (57.8 for nonvegetarians and 58.1 for vegetarians) and the majority (>85%) were less than 65 years of age (Table 1).

There are several possible mechanisms for our findings:

### 4.1. Cardiometabolic Risk Factors

Taiwanese vegetarian diets, like other plant-based diets, improve weight and glycemic control, diabetes, decrease blood lipids and blood pressure and reverse atherosclerosisand reduce the incidence of strokes and depression [8,9,10,11]. All these cardiometabolic risk factors in midlife increase amyloid deposition and dementia in late life [2,16,17,18]. However, they do not seem to affect the progression of AD in later life [19]. As a matter of fact, hypertension appeared to have protective effect against dementia in older population [20]. Thus, the results of our study showing that the relatively young age of our study population (midlife) conferred a protective effect against dementia in later life is in accordance with the findings of current medical literature.

### 4.2. Inflammation 

Neuroinflammation has been demonstrated to be a major etiological factor in the pathogenesis and progression of AD [21]. Available evidence indicates that consumption of Mg, fiber, polyunsaturated fatty acids, flavonoids, and carotenoids (which are commonly found in vegetarian diets) is associated with decreased blood levels of tumor necrosis factor alpha, high-sensitivity C-reactive protein, and interleukin-6, known inflammatory markers [22]. In fact, in epidemiological studies, plant-based diets such as Mediterranean, Dietary Approach to Stop Hypertension, have been demonstrated to protect against dementia in old individuals [23,24]. In addition, arachidonic acid (mainly from meat) had been demonstrated to cause a cascade of neuroinflammation [25]. Previous studies showed that blood arachidonic acid (AA) was higher in omnivores than in vegetarians because meat, poultry, and fish are the major sources of AA in the diet [26,27]; thus, vegetarian diets appeared to protect against inflammation and, hence, dementia.

### 4.3. Gut Microbiota (GM) 

Through the vagus nerve, GM mediates or modulates various central nervous system processes and produce metabolites and immune mediators that cause changes in neuroinflammation, neurotransmission, and human behavior. In a study with AD patients, major changes of the gut microbiota were demonstrated [28]. Pathogenic and commensal enteric bacteria affect immune system and brain function through the production of lipopolysaccharides and amyloid [29]. GM dysbiosis plays an important role in the development of AD, by causing amyloid-beta aggregation and neuroinflammation [30,31]. GM produces catecholamine, serotonin, kynurenine, etc., which modulate brain functions [32]. It is postulated that dementia starts with the GM dysbiosis, causing systemic and local inflammation, and derangement of the gut–brain axis [32]. On the other hand, people on plant-based diets have stable and more diverse microbial systems with less dysbiosis, which is beneficial for human health [33]. Further study is required to delineate the exact underlying mechanisms by which plant-based diets benefit dementia via the microbiota–gut–brain axis.

### 4.4. Dietary Pattern 

Recent literature indicated that many individual nutrients or phytochemicals improve cognitive function [34,35]; however, to study the health effects of culturally, ethnically, and geographically diverse dietary practices remains a big challenge. The dietary patterns across different geographic areas, ethnicities, and cultural habits worldwide are quite different in their composition and contents of macronutrients and micronutrients, polyphenols, and other phytochemicals. Taiwanese diets (both omnivores and vegetarian) are similar to traditional Asian or Chinese diets but quite different from the typical Western diets in that they include more soy, rice, wheat, and salt but less meat and milk products (cheese) [9]. Meat consumption has been demonstrated to increase the prevalence of dementia. In Japan, the prevalence of AD increased several folds from 1985 to 2008 when the dietary habits changed gradually from the traditional Japanese diet to the diet with higher meat content, typical in Western countries [36]. In addition, it had been shown in the study of ten countries that the higher the national dietary meat supply, the higher the prevalence of AD [37]. In cohort studies, processed meat has been demonstrated to increase the incidence of all-cause dementia [38] and plant-based dietary pattern such as vegetarian and Mediterranean appeared to associate with reduced rates of cognitive impairment, AD, and dementia [5,39,40,41]. However, the effect of meat consumption on dementia remained inconclusive. A systemic review of 29 studies demonstrated that the majority showed no correlation between meat consumption and cognitive function or disorders [42]. Animal meats contain high saturated fat, total fat, and cholesterol, which are considered detrimental to human health, but on the other hand, provide essential nutrients (protein, essential amino acids, vitamins, and minerals), which are beneficial to human health. Various studies of the association between meat consumption and the incidence of dementia probably depend on the interplay of these conflicting factors. In our study, both the study and control groups stayed on similar diets, except the study group omitted foods of animal origin, such as poultry, fish, and meat, due to religious reasons. In addition, our vegetarians and nonvegetarians were comparable in their health and socioeconomic status, the majority did not smoke cigarettes or drink alcohol. This allowed us to study the unique contribution of meat to the incidence of dementia, excluding most of other potentially confounding factors. Hence, our study provided evidence that, compared with nonvegetarians, Taiwanese vegetarians (meatless) in midlife are associated with reduced incidence of clinically overt dementia in later years. This is a valuable addition to the vast amount of knowledge regarding the association of diets with the incidence of dementia and helps in the policy development of strategies for the prevention of dementia.

The strengths of the study are as follows: (1) vegetarians in this study were Buddhists from the Tzu Chi Foundation, which ensures the high adherence to a vegetarian diet. (2) Our study is the first prospective one to study the association of midlife long-term vegetarian dietary pattern with the risk of dementia in later years. (3). The NHI covers nearly 100% of the population in Taiwan, so that recall bias of comorbidity and loss to follow-up were minimized.

This study has several limitations: (1) Only those with clinically overt dementia that required medical attention were included in our study, those with minor cognitive impairment were not studied. (2) Dietary pattern was only assessed at baseline, and changes over time could still be possible. (3) Dementia cases were based on the NHIRD medical claims records, defined by ICD-9 diagnosis codes, so that the causes of dementia might not be accurate and different subtypes of dementia could not be adequately evaluated. (4) Selection bias cannot be excluded. Those who select vegetarian diets may have less tendency to have dementia in the first place. (5) Due to the design of our study and despite adjustments for potential confounding factors, residual confounding may still be present. (6) This is a study of Taiwanese vegetarians, and generalization to other populations requires further study.

## Figures and Tables

**Figure 1 nutrients-14-00588-f001:**
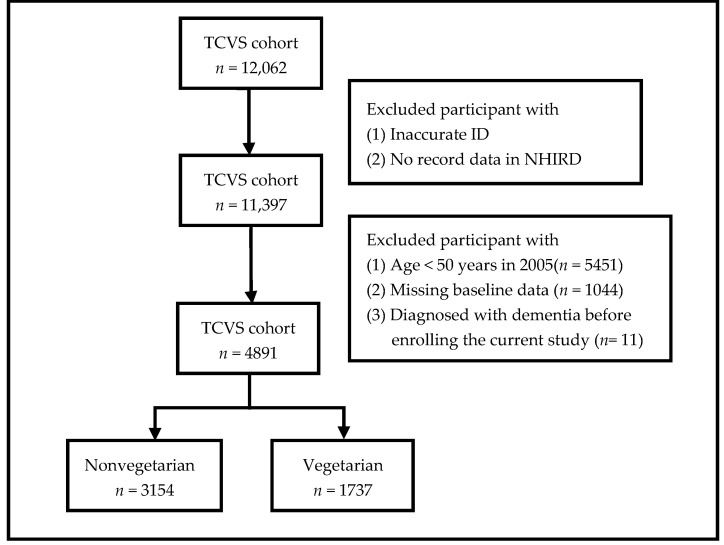
Flowchart of The Study. Abbreviations: TCVS, Tzu Chi Vegetarian Study; ID, the unique national identification number; NHIRD, National Health Insurance Research Database.

**Figure 2 nutrients-14-00588-f002:**
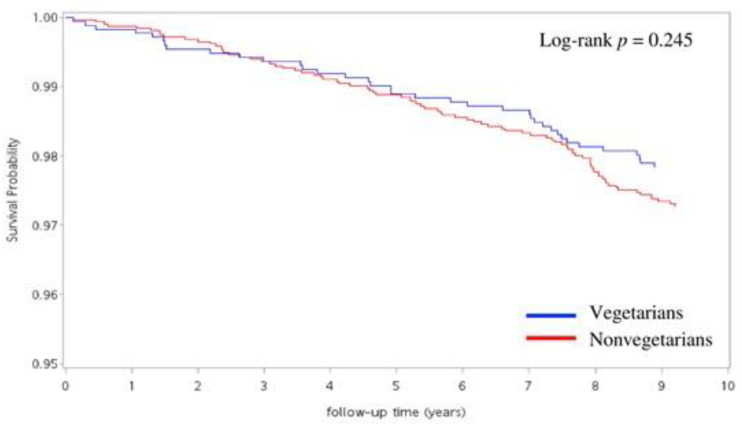
Cumulative incidence of dementia between vegetarians and nonvegetarians during a 10-year follow-up period.

**Table 1 nutrients-14-00588-t001:** Sociodemographic characteristics of vegetarians and nonvegetarians.

	Nonvegetarians	Vegetarians	*p*-Value
	*n* (%)	*n* (%)	
Total	3154 (64.5)	1737 (35.5)	
Sociodemographic characteristics			
Age, Mean (SD)	57.8 (6.3)	58.1 (6.5)	0.253
Age group			
<65 years	2704 (85.7)	1477 (85.0)	0.525
65–74 years	408 (12.9)	221 (12.7)	
≥75 years	42 (1.3)	39 (2.2)	
Sex			<0.01
Male	1345 (42.6)	461 (26.5)	
Female	1809 (57.4)	1276 (73.5)	
Education level			<0.01
≤Elementary school	1008(32.0)	658 (37.9)	
Middle and high school	1421 (45.1)	744 (42.8)	
Higher education	725 (23.0)	335 (19.3)	
Marriage			<0.01
Married	2940 (93.2)	1576 (90.7)	
Single/Divorced/Widowed	214 (6.8)	161 (9.3)	
Lifestyle habits			
Regular exercise habit	1470 (46.6)	689 (39.7)	<0.01
Smoking	529 (16.8)	173 (10.0)	<0.01
Alcohol drinking	507 (16.1)	199 (11.5)	<0.01

**Table 2 nutrients-14-00588-t002:** Baseline medical comorbidity of vegetarians and nonvegetarians.

	Nonvegetarians	Vegetarians	*p*-Value
	*n* (%)	*n* (%)	
Hypertension	554 (17.6%)	282 (16.2%)	0.250
Diabetes mellitus	449 (14.2%)	190 (10.9%)	0.001 *
Hyperlipidemia	621 (19.7%)	310 (17.8%)	0.119
COPD	286 (9.1%)	163 (9.4%)	0.717
Hyperthyroidism	70 (2.2%)	55 (3.2%)	0.047 *
Hypothyroidism	61 (1.9%)	46 (2.6%)	0.103
Cerebrovascular disease	393 (12.5%)	172 (9.9%)	0.008 *
Ischemic Heart disease	454 (14.4%)	221 (12.7%)	0.109
Cardiac arrhythmia	290 (9.2%)	198 (11.4%)	0.015 *
Renal disease	181 (5.7%)	82 (4.7%)	0.145
Anxiety disease	396 (12.6%)	228 (13.1%)	0.591
Major depressive disorder	69 (2.2%)	31 (1.8%)	0.398
Substance use disorder	15 (0.5%)	3 (0.2%)	0.041 *

Abbreviation: COPD, chronic obstructive pulmonary disease. * *p* < 0.05.

**Table 3 nutrients-14-00588-t003:** Stepwise hazard ratio estimation of dementia risk in vegetarian compared with nonvegetarian.

	Nonvegetarians	Vegetarians
Cases/Person-year	84/28,798.47	37/15,926.9
Crude, HR (95% CI)	1	0.796 (0.540–1.171)
Adjusted for age, sex	1	0.688 (0.466– 1.018)
Adjusted for age, sex, education level, marriage	1	0.676 (0.457–1.001)
Adjusted for age, sex, education level, marriage, regular exercise, smoking, drinking	1	0.661 (0.446–0.979) *
Adjusted for age, sex, education level, marriage, regular exercise, smoking, drinking, baseline medical comorbidity	1	0.671 (0.452– 0.996) *

Abbreviations: HR, hazard ratio; CI, confidence interval. * *p* < 0.05.

**Table 4 nutrients-14-00588-t004:** Risk of developing dementia in vegetarians compared with nonvegetarians in age subgroups.

	Nonvegetarians		Vegetarians		Adjusted HR *	95% CI	*p*-Value
	N	Person-Years	N	Person-Years			
**Stratification of Age Groups**							
50–64 years	36	24,906.05	12	13,678.64	0.601	0.309–1.166	0.422
65–74 years	39	3560.74	15	1971.74	0.613	0.333–1.129	0.794
≥75 years	9	331.69	10	276.51	1.089	0.335–3.543	0.233

Abbreviations: HR, hazard ratio; CI, confidence interval. * After adjusting for sex, education level, marriage, regular exercise, smoking, drinking, and medical comorbidity.

## Data Availability

The data used to support of this study are included within the article.

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
