# Peer review of "Taiwanese Vegetarians Are Associated with Lower Dementia Risk: A Prospective Cohort Study"

_nutrients, 2022, doi:10.3390/nu14030588_

Round 1

Reviewer 1 Report

This article aimed to explore the difference of dementia incidence between vegetarians and nonvegetarians in the TCVS population during a ten-year period. There are some questions:

  1. The definition of vegetarian used in this article is not a strict definition, eggs, milk and other animal foods were not considered;
  2. After the baseline questionnaire survey in 2005, how often was the follow-up conducted in the middle term? What kind of follow-up was used? Whether there was no follow-up in the middle term, and only matched the National Death Registry and the National Health Insurance Research Database at the Health and Welfare Data Science Center at Ministry of Health of Taiwan at the end to obtain outcome information. How to assess whether the subjects' dietary patterns changed during the 10-year period(2005-2014)?
  3. By using the information of the National Health Insurance Research Database that only records the patient's proactive visits, were there more dementia patients who have not yet seen doctors and not counted?
  4. Should the possible impact of the nutritional status of the two groups of subjects on the research results be considered?
  5. The possible mechanisms analysis in discussion part has poor correlation with the content of this study;
  6. Some punctuation marks are inappropriate;
  7. Please provide the ethical approval number, if possible;
  8. Line 36 TCHS? Line 160 and 171: Table 4?

Reviewer 2 Report

This study shows the potential efficacy of Taiwanese vegetarian diet in the decreased risk of incident dementia. The topic and the results are interesting mainly because to date it is still difficult to identify specific efficacious treatments for AD; and a non-pharmacological therapeutic treatment could represent a valid alternative to the classical pharmacological approach. As far as I understand, the methods are sound. There is a limitations section.

Thus, I have just some minor points to address:

a) check for acronymous: for instance, AD and then Alzheimer disease (AD) (page 6 line 185 and line 192);

b) check for English mistakes or typos : for instance, “dementia start” (line 212 page 7); “On the other hand, People on..” (line 214 page 7); “addition, It had been shown…” (line 230 page 7)

Round 2

Reviewer 1 Report

There is still not enough conclusive information on how to access  whether the dietary patterns of the subjects have changed in the past 10 years? How many patients with dementia have not been recorded because they did not take the initiative to see a doctor? and it is suggested that the author should compare whether there are differences in nutrient intake between the two groups.
